

**Peak river flows in cold regions – Drivers and modelling using GRACE satellite**
**observations and temperature data**
Shusen Wang[1,*], Fuqun Zhou[1], Hazen A. J. Russell[2], Ran Huang[3], Yanjun Shen[4]
[1] Canada Centre for Remote Sensing, Natural Resources Canada, Ottawa, ON, Canada
[2] Geological Survey of Canada, Natural Resources Canada, Ottawa, ON, Canada
[3] College of Information & Electrical Engineering, China Agricultural University, Beijing, China
[4] Key Laboratory of Agricultural Water Resources, Chinese Academy of Sciences, Shijiazhuang,
China
**Running title: Forecasting peak river flows in cold regions**
[*]Corresponding author:
560 Rochester Street, Ottawa, ON K1A 0E4, Canada
Phone: (613) 759-6462
E-mail: Shusen.Wang@Canada.ca





**Abstract**

The peak river flow for the Mackenzie River is modelled using GRACE satellite
observations and temperature data, which advances the applications of space-based time-variable
gravity measurements in cold region flood forecasting. The model estimates peak river flow by
simulating peak surface runoff from snowmelt and the corresponding baseflow. The modelled
results compared fairly well with the observed values at a downstream hydrometric station. The
results also revealed an average 22-day travel time for the snowmelt water to reach the
hydrometric station. The major driver for determining the peak flow was found to be the
temperature variations. Compared with the Red River basin, the results showed that the
Mackenzie River basin has relatively high water storage and water discharge capability, and low
snowmelt efficiency per unit temperature. The study also provides a GRACE-based approach for
basin-scale snowfall estimation, which is independent of in situ measurements and largely
eliminates the limitations and uncertainties with traditional approaches. The model is relatively
simple and only needs GRACE and temperature observations for peak flow or flood forecasting.
The model can be readily applied to other cold region basins, and could be particularly useful for
regions with minimal data.

**Key Points:**

• Peak runoff and winter baseflow are modelled using GRACE observations.
• Major drivers controlling peak river flows or floods in cold regions are identified.
• Mechanisms underlying peak flow regimes of different basins are discussed.
• GRACE observation is used for basin-scale snowfall estimate.



## 1. Introduction

Peak river flow and its forecasting are of considerable interest to flood control and
emergency service agencies, river recreationists, wildlife managers, hydropower plant operators
and anyone interested in the combined effect of watershed yield and human regulation on the
river maximum flows at a site. Flooding is a hazard to developed areas and to human activities in
the floodplain and causes human hardship and economic loss (International Joint Commission.
2000; Burton et al., 2016). Flooding ranks as one of the most damaging forms of natural disasters
in the world. On the other hand, flooding is essential to a healthy environment. Floods can
benefit the natural environment and sustain many ecosystems. For instance, the recurrent
inundation of the Peace-Athabasca Delta in the Mackenzie River basin in Canada has fostered an
environment in which plant and animal life has achieved a balance dependent on flooding. In
fact, water spilling across the floodplain nourishes the wetlands of many of Canada's major
deltas (Andrews, 1993).

Peak river flow in cold regions is commonly a result of snowmelt during the spring
break-up, or the subarctic nival regime (Church, 1974). During the winter most of the
precipitation is in the form of snow, which accumulates in the basins until the spring breakup.
Depending on the temperature in the spring freshet, huge quantities of water from snowmelt can
be produced. Meltwater is unable to penetrate when the ground is frozen and runs off over the
ground surface into rivers and lakes, resulting in peak flows and floods. Snowmelt runoff floods
are the most common type of flooding in Canada (Andrews, 1993; Yang et al., 2009; Rannie,
2015; Wang and Russell, 2016).

Two key factors determining the magnitude of peak river flows or floods during the
freshet are the amount of snow water equivalent (SWE) available at the time of spring breakup





and the temporal patterns of rising air temperature during the snowmelt season. As such,
forecasting peak flows or floods for rivers in cold regions, regardless of the methods being used,
heavily relies on information for basin-scale SWE and temperature. SWE can be estimated by
the difference of snowfall and water loss from snow sublimation. Unfortunately, accurately
measuring snowfall is difficult and often involves large uncertainties. Studies show that the
measurement errors of snowfall at climate stations can be as high as 50% due to the wind-
induced undercatch and the fact that many trace events cannot be easily measured (Goodison et
al., 1998). Snow sublimation estimation is also difficult and has large uncertainties (Wang et al.,
2013; Wang et al., 2015a). In addition, cold regions commonly have very sparse observational
data. For instance, the density of climate stations in the Canadian Northwest Territory is only
about 2 stations per 100,000 $km^2$ (Metcalfe, 1994). This sparse spatial density of stations makes
the up-scaling of SWE from site to basin-scale extremely difficult and unreliable. Remote
sensing, either by optical (e.g., Hall et al., 2002) or microwave (e.g., Dong et al., 2005) satellite
sensors, can provide valuable information for the spatial snow cover distribution. However,
remote sensing approaches for the estimation of SWE depend on retrieving snow parameters
such as snow cover area, snowpack depth, density, liquid and ice content, etc., which can be
significantly affected by the climate/atmosphere and land surface/snowpack conditions, such as
the temperature effect on snow density, and cloud, vegetation cover and topographic impacts on
snow cover area identification (Nolin, 2010). Moreover, SWE models from remote sensing
heavily rely on in situ data for calibration and validation, which may propagate the errors in the
in situ measurements to the remote sensing products (Hall and Riggs, 2007). Due to the
difficulties mentioned above, spatial snow data products either from traditional methods or from
remote sensing approaches often have large uncertainties. For instance, recent studies have found



that errors in snow data are the principal contribution to the water budget imbalance in cold-
region basins (Wang et al., 2014a; Wang et al., 2014b; Wang et al., 2015b). Consequently,
accurate estimation of basin-scale SWE is a key challenge in improving flood forecasting over
cold region rivers.
In a recent study, Wang and Russell (2016) developed a flood forecasting model using
the Gravity Recovery and Climate Experiment (GRACE) satellite observations and temperature
data. In the model, the total water storage (TWS) data from GRACE was used to estimate the
basin-scale SWE at the spring breakup. The model does not require snowfall and sublimation
data as well as the site-to-basin upscaling process, thus largely eliminating the data constrains
discussed above. In Wang and Russell (2016), the model was applied to the Red River basin, a
USA-Canada transboundary watershed located in central North America, mostly in the U.S.
states of Minnesota and North Dakota. Floods predicted by the model compared well with the
observed values. However, the Red River has a drainage area of 116,500 km², which is limited in
size relative to the large footprint of the GRACE satellites. This size limitation may contribute to
the large data uncertainties in the TWS of the basin which might substantially impact the
accuracy of flood forecasts. Second, a conclusion found in Wang and Russell (2016) is that the
SWE at the spring breakup is the major driver determining the magnitudes of peak river flows,
with temperature playing the secondary role. Whether this conclusion is valid for other basins
with different hydroclimatic conditions still remains to be investigated.
In this study, the model of Wang and Russell (2016) is calibrated to forecast the peak
flows for the Mackenzie River in Canada (Fig. 1). The Mackenzie River basin has a large
drainage area of 1.8 million square kilometres, which largely eliminates the constraints imposed
by the large footprint of GRACE. The basin is located in northwest Canada, about 18 degrees of





latitude north of the Red River basin. Compared with the Red River basin, the Mackenzie River
basin has very different physiographic and hydroclimatic conditions, including the very low
temperature, large amount but small interannual variations in winter snowfall, as well as high
water storage conditions. As a result, the Mackenzie River has much delayed peak discharge
flows that have much smaller interannual variations than the Red River. The objective of this
study is to evaluate the model performance in forecasting, and to identify the major drivers in
determining, the peak flows or floods for the Mackenzie River. The model results and parameter
values will be compared with that for the Red River to help better understand the mechanisms
underlying the peak river flows for basins with different physiographic and hydroclimatic
conditions.
**2. Study Region, Data, and Hydroclimate Characterization**

The Mackenzie River Basin is one of the largest North American river basins and covers

about 1.8 million square kilometres or about 20% of the landmass of Canada. Our basin study
area is located between 52° to 70° N and encompasses three major physiographical regions. In
the west, the Western Cordillera consists of a series of mountain chains and valleys or high
plateaus. Many ridges of the Rocky Mountain chain exceed 2000 m elevation, and some have
glaciers occupying the mountain tops and high valleys. To the east is the Canadian Shield, a
rolling terrain with myriad of lakes and valley-wetlands separating upland outcrops of
Precambrian bedrock. The central zone is part of the Interior Plains, with wetlands, lakes, and
vegetation that ranges from prairie grassland in the south, through the boreal and subarctic
forests, to the tundra in the north (Woo and Thorne, 2003). There are a large number of lakes
across the basin, with the top three large lakes (i.e., Great Bear Lake, Great Slave Lake and Lake
Athabasca) having a total surface area of over 67,000 $km^2$. The basin has a wetland coverage of



approximately 49% and permafrost underlies approximately 75%. The Mackenzie River system
flows a total of 4,241 kilometres from its headwaters northwards to the Arctic Ocean.

The data required by the model for peak flow estimation includes GRACE TWS and

daily temperature $T_a$. The in situ measurement of river flow $Q$ is used for model calibration. The
GRACE Release-05 TWS datasets were downloaded from the GRACE Tellus website
(ftp://podaac-ftp.jpl.nasa.gov/allData/tellus/L3/land_mass/RL05/) (Swenson. 2012). The datasets
include monthly TWS from three data processing centers: CSR (University of Texas/Center for
Space Research), GFZ (GeoForschungsZentrum Potsdam), and JPL (Jet Propulsion Laboratory).
The data are provided in a spatial sampling of 1 degree grids in both latitude and longitude. The
land grid scale factor, as provided with the TWS data, was applied to recover signal loss due to
filtering and truncation (Swenson and Wahr, 2006). Monthly error estimates in the GRACE
TWS for the Mackenzie River Basin are based on combined measurement and leakage error,
following Wahr et al. (2006). A detailed description of the data processing and accuracy
assessment can be found in Landerer and Swenson (2012). The differences among the three
datasets, as shown later, were found to be small over our study region. In this study, the average
TWS of the three datasets was used. The study covers 12 snow-years from the fall of 2002 to the
spring of 2014. The baseline of the TWS, which was based on the average from January 2004
through December 2009 in the original data, was re-adjusted to the minimum value found over
the 12-year period.

The basin average $T_a$ was calculated from the Global Land Data Assimilation System

(GLDAS) meteorological forcing of temperature, which is provided at a 3 hour time step and a
spatial resolution of 0.25×0.25 degree latitude/longitude. The air temperature time series for
2002-2014 was downloaded from Goddard Earth Sciences Data and Information Service Center





(http://mirador.gsfc.nasa.gov/). The daily air temperature of the grids is taken as the average of
the 8 readings of the day, and the daily $T_a$ of the basin is the average of the daily values of all the
GLDAS grids within the basin. The GLDAS precipitation is also processed to assist our
discussion. The daily precipitation of each grid is the sum of the 8 readings of the day, and the
daily value of the basin is the average of all the grids within the basin.

The in situ daily $Q$ observed at the hydrometric station of Mackenzie River at Arctic Red

River (station ID: 10LC014, Latitude: 67° 27' 21" N, Longitude: 133° 45' 11" W) was obtained
from the Water Survey of Canada (http://www.ec.gc.ca/rhc-wsc/). The $Q$ collected at this
location, before the river branches into many distributaries, is considered as the total flow for the
Mackenzie River system. The station controls an area of 1,679,100 km² (> 90% of the basin).
The original $Q$ is in m$^3$ s$^{-1}$ and it was converted to water depth (mm) using the basin area. The
peak flow is the maximum mean daily flow (the highest average flow for an entire day) in a year
at the station. It is worth noting that the $Q$ data is only required for model calibration. The peak
flow estimation, once the model is calibrated, will only need TWS and $T_a$ prior to winter and at
the snowmelt season in spring.

The Mackenzie River Basin has several climatic regions, including cold temperate,

mountain, subarctic, and arctic zones. According to the climate data used in this study, the daily
$T_a$ averaged for the basin ranged from a low of -36 ℃ to a high of 21 ℃ during the 12 study
years (Fig. 2). The 12-year average $T_a$ of the basin varied from -22.6 ℃ in winter to 16.4 ℃ in
summer, with an annual mean of -2.4 ℃. The temperature dropped below 0 ℃ in mid-October
and rises above 0 ℃ at the end of April, with an annual average of more than 200 days having
temperatures below 0 ℃. In winter, the basin is in a deep frozen state and precipitation as
snowfall accumulates until spring breakup. Annual precipitation over the 12 years ranged from a





low of 311 mm to a high of 432 mm, with an average of 386 mm (Fig. 2). Annual precipitation
during the winter time when $T_a$<0 ºC ranged from 124 mm to 189 mm, of which the average was
157 mm, or 41% of the total annual precipitation. However, it is generally recognized that
snowfall from the climate station measurements is underestimated. This will be further discussed
using our GRACE-based estimates in Section 4.

The TWS of the basin has strong seasonal variations, with highest values around April

before snow breakup and lowest values around October before snow starts to accumulate (Fig.
2). The average seasonal variation range (peak to peak) for the 12 years was 116 mm. The
pronounced seasonal pattern is mainly a result of snow accumulation combined with extremely
low evapotranspiration and river discharge in winter, and high evapotranspiration combined with
high river discharge in summer (Wang and Li, 2016). The TWS has moderate interannual
variations. Among the 12 years, the maximum differences reached 68 mm in April and 77 mm in
October. The three original TWS datasets have an average difference of 5.4 mm, which is minor
compared with the seasonal variations of TWS for the basin. The combined measurement and
leakage error for the GRACE TWS datasets is estimated at 9.9 mm, which is about 8.5 % of the
average seasonal variation of TWS.

River flow of the Mackenzie River is greatest in spring when snowmelt occurs (Fig. 2).

The peak flows for the study period observed at the station 10LC014 varied from a low of 24,200
$m^3$ $s^{-1}$ in 2010 to a high of 35,000 $m^3$ $s^{-1}$ in 2013 (Fig. 2). The 12-year average peak flow was
28,400 $m^3$ $s^{-1}$. The date when peak river flow occurred was on June 3 on average, with the
earliest on May 15 in 2005 and latest on July 12 in 2007. In the climate change context, studies
have showed that the Mackenzie River is experiencing a shift of peak flows to earlier in the
spring due to earlier melt of snow cover and river ice breakup (Prowse et al., 2010; Woo and


Thorne, 2003; Yang et al., 2014). The basin discharge in winter gradually decreases and at the
time of spring breakup, the observed 12-year average flow was around 3,700 $m^3$ $s^{-1}$. In summer,
although the total amount of rain was higher than the winter snowfall, the river discharge peaks
were significantly lower (Fig. 2), mainly due to the high evapotranspiration, recharge of water to
the soil and aquifers, and relatively even distribution of rain over the season. The large lakes in
the basin also provide natural regulation to the system.
**3. Methods**
**3.1. Model Description**

The model includes five major steps that are summarised below.

**Step 1: Determining snow season and the total water storage change**

This study only concerns the snow season, defined as the time period from the start ($t_0$) of

snow accumulation in late autumn to the snow breakup ($t_b$) in the next spring (Fig. 3). The $t_0$ is
determined by the criteria of (1) the first precipitation event in late fall when daily average
temperature $T_a$ drops below 0°C and (2) the accumulated $T_a$ after this date remains negative. The
criteria are to ensure that precipitation is in the form of snow and to exclude temporary snowfalls
that are likely to melt in early winter. After this date, the net snow amount, i.e., snowfall minus
snow sublimation, is accumulated over the basin in winter during which the $T_a$ remains far below
0°C (Fig. 2). The spring breakup time $t_b$ is determined by the criteria of (1) daily average
temperature $T_a$ rises above 0°C and (2) the accumulated $T_a$ after this date remains positive. The
criteria are to ensure that the heat conditions of the basin will lead to the spring breakup and to
exclude the possible minor snowmelt events in the snow cover season in case the $T_a$ momentarily
rises above 0°C.



The basin total water storages at $t_0$ and $t_b$, i.e., $TWS_0$ and $TWS_b$, are estimated by linearly
interpolating the GRACE TWS for the two months before and after $t_0$ and $t_b$, respectively. The
$TWS_0$ represents the maximum amount of non-snow water in the basin during the snow cover
season. It mainly consists of surface water, soil water, and groundwater, part of which will be
discharged in winter. The $TWS_b$ represents the sum of net snow accumulation ($S_b$) and non-snow
water content left in the basin at time $t_b$ (Fig. 3).
**Step 2: Modelling baseflow**
In winter due to the frozen soil and snow cover, water infiltration and evaporation at the
soil surface is minimal. The decrease of non-snow water ($W$) in the basin, $dW(t)/dt$, is thus
mainly due to basin discharge or baseflow. In this study, the winter baseflow $Q_{base}$ is modelled
using a first order differential equation:
$$Q_{base}(t) = -dW(t)/dt = a(W(t)-b) \tag{1}$$
where $a$ is a parameter representing the lump conductivity of the basin for water discharge, and $b$
is a parameter representing the threshold value of $W$ at which the basin would have zero
discharge. This simplified model represents the basin water discharge as proportional to the
available water storage and lump basin conductivity for discharge.
With the above model and the initial condition of $W(t_0)=TWS_0$, the accumulated total
baseflow in winter, $Q_{sum}$, can be determined by:
$$Q_{sum} = \int_{t_0}^{t_b} Q_{base}(t)dt = TWS_0 - (TWS_0 - b)e^{-a(t_b - t_0)} - b \tag{2}$$





The values for parameters $a$ and $b$ are obtained numerically by finding the least square errors

between the observed and modelled $Q_{sum}$ for the 12 study years. Once $a$ and $b$ are known, the

baseflow rate at any time from $t_0$ can be calculated as:

$$Q_{base}(t) = a(TWS_0 - b)e^{-at}. \qquad (3)$$

**Step 3: Determining snow water equivalent at spring break-up time**

After $Q_{sum}$ (Eq. 2) is known, the snow water equivalent at $t_b$, $S_b$, can then be obtained as

the sum of $Q_{sum}$ and the change of TWS in the snow season (i.e., $TWS_b - TWS_0$) based on water

balance of the basin (Fig. 3):

$$S_b = TWS_b - TWS_0 + Q_{sum} \qquad (4)$$

The $S_b$ represents the initial amount of snow which is going to melt following the

temporal trajectory of air temperature as described next. Note that snow sublimation in winter is

implicitly included in the estimate of $S_b$ as GRACE measures the net change of water storage,

which is contributed by the difference of snowfall and snow sublimation in winter.

**Step 4: Modelling snowmelt ($M$) and peak surface runoff ($Q_{runoff}$)**

The snowmelt is estimated by a temperature index model at a daily time step:

$$M(t) = \min(S(t), \alpha(T_a(t) - \beta)) \qquad (5)$$

where $M(t)$ and $S(t)$ is the snowmelt and snow amount available on day $t$ (Note $S(t)=S(t-1)-M(t-1)$), respectively, $\beta$ is a base temperature for snowmelt and $\alpha$ is the snowmelt rate per unit of

temperature above $\beta$. Equation (5) determines the actual snowmelt rate by both temperature and

snow availability on a given date. The time series of snowmelt is calculated using the initial



259 condition of $S_b$ (Fig. 3) and the daily $T_a$ time series. The parameter values for $\alpha$ and $\beta$ were

260 solved using a nested numerical iteration scheme to find the best correlation between peak

261 snowmelt rate and the observed peak surface runoff. Observed peak surface runoff is calculated

262 as the difference between observed peak river flow and the corresponding baseflow obtained

263 using Equation (3). The model for estimating peak surface runoff, $Q_{runoff}$, from peak snowmelt is

264 obtained after this numerical process is done.

265 **Step 5: Modelling peak river flow**

266   The modelled peak river flow is the sum of peak surface runoff and the corresponding

267 baseflow obtained above:

$$Q_{peak} = Q_{base} + Q_{runoff} \tag{6}$$

269   In summary, the model determines peak river flows by simulating its two components of

270 peak surface runoff from snowmelt and the corresponding baseflow from basin discharge. The

271 snowmelt is simulated by a temperature index model.  The basin discharge is simulated by a first

272 order differential equation model. There is a total of four parameters, $a$ and $b$, which are

273 calibrated using the total winter flow, and $\alpha$ and $\beta$, which are calibrated using the observed peak

274 surface runoff. Once calibrated, the model only needs GRACE TWS and $T_a$ as inputs for

275 determining peak river flow after spring breakup.

276 **3.2. Model Evaluation**

277   The model results were compared to in situ $Q$ observations and evaluated using mean

278 absolute error ($MAE$), the Pearson correlation coefficient ($r$) and t-test for significance levels ($p$),

279 as well as the Nash–Sutcliffe model efficiency coefficient ($E$). The $E$ is commonly used to assess





the predictive power of hydrological models. The leave-one-out cross-validation (LOO-CV)
approach is used to evaluate the general model performance in peak river flow forecasting. The
result from this LOO-CV evaluation is generally a more conservative estimate of the model
performance than that trained on all samples. The model results were also compared with other
basins to understand the mechanisms underlying the variations of peak flows over different
regions.

**4. Results and Discussion**

The model gives an average estimate for the start of snow cover of $t_0$ on October 14 and

spring breakup of $t_b$ on April 29. The average snow water equivalent accumulated during this
time period (197 days) is $S_b$=160 mm. Given a water loss of 22 mm due to snow sublimation
based on Wang et al. (2015a), the total snowfall during this time period from $t_0$ to $t_b$ is thus 182
mm. In comparison, the corresponding total precipitation amount from the GLDAS datasets as
discussed in Section 2 is 150 mm (snowfall=133 mm, rain=17 mm), which is about 20% lower
than the GRACE-based estimate. Underestimation in snowfall over cold regions has been
recognised by many studies (e.g., Wang et al., 2014a; Wang et al., 2015b) due to the errors as
discussed in the Introduction section. This is particularly so for our study region as the available
benchmark measurements, while extremely sparse, are mostly located in valleys or lowlands
where the data are likely to underestimate the average condition of the entire basin. Many efforts
have been made to correct the biases (e.g., Woo et al., 1983; Yang et al., 2005) but they are
largely constrained by the difficulties in quantifying snow amounts at basin-scale. Our study
provides a new GRACE-based approach for estimating snowfall at basin-scale. The approach is
largely independent of in situ snowfall data thus eliminates most of the limitations and
uncertainties in the snow gauge measurements and the site-to-basin up-scaling processes.





303 The model results show that the basin has a lump conductivity of $a=1.07\times10^{-3}$ day$^{-1}$ for

304 water discharge, and a threshold value of $b=-195.9$ mm of water below which the basin would

305 have no discharge (Table 1). Compared with the Red River basin (Wang and Russell, 2016)

306 which has $a=0.49\times10^{-3}$ day$^{-1}$ and $b=9.2$ mm, the model results suggest that the Mackenzie River

307 basin has a relatively high basin conductivity and high water storage state. This is consistent with

308 the facts that the Mackenzie River basin has about 49% of its area as wetland and a large number

309 of lakes which are highly effective in providing large storage capacities and are available for

310 water discharge. Annual evapotranspiration for the basin is less than 60% of its annual

311 precipitation (Wang et al., 2015a), so the basin has a large water surplus for soil and aquifer

312 recharge as well as lake and wetland replenishment to sustain the winter low flows. In contrast,

313 the Red River basin has a very flat terrain (the slope of the river averages less than 10 cm per

314 kilometer). It also has high evapotranspiration that is close to its precipitation in summer,

315 resulting in a very low water storage state prior to winter. The model results are also agreeable

316 with the observed difference in winter flows, which are as low as 5.8 mm day$^{-1}$ for the Red River

317 and as high as 48.8 mm day$^{-1}$ for the Mackenzie River as discussed next.

318 The model performance for winter baseflow estimation is shown in Table 1 and Fig. 4A.

319 Compared to the mean observed baseflow of 48.8 mm in winter over the study period, the model

320 has a mean absolute error of $MAE=2.37$ mm, or 4.9% of the observed value. The modelled

321 winter baseflow for the 12 years has a correlation coefficient of $r=0.73$ with the observed values

322 at a significance level of $p<0.007$. The Nash–Sutcliffe model efficiency coefficient for baseflow

323 estimates is $E=0.53$. The results suggest that the basin winter discharge is primarily driven by its

324 water storage prior to winter. Indeed, surface runoff is minimal in winter due to the lack of liquid

325 precipitation. Water exchange between soil and groundwater is also minimal due to the frozen




soil. River flow in winter is thus sustained by groundwater and lake discharge that is controlled
by pre-winter storage conditions. Determining groundwater and lake water storage at the basin
scale is extremely difficult by traditional methods. The above results underscore the advantages
of using GRACE satellite observations in estimating basin water storages and discharge in
winter.

The snowmelt model suggests a snowmelt rate of $\alpha$=17.0 mm per unit of temperature

above a base temperature of $\beta$=2.1°C (Table 2) for the Mackenzie River basin. The results are
close to that obtained for the Red River basin which has $\alpha$=18.2 mm °C$^{-1}$ and $\beta$=1.0°C (Wang
and Russell, 2016). The slightly higher melting power for a unit temperature with a lower base
temperature for the Red River basin reflects the impacts of other environmental variables on the
snowmelt, such as higher solar radiation for the Red River basin than for the Mackenzie River
basin. Comprehensive and physically based snowmelt models are available and they have the
advantages of simulating the integrated impact of all environmental variables (e.g., radiation,
humidity, wind speed) on the snowmelt processes (e.g., Wang et al., 2007; Wang, 2008; Zhang et
al., 2008), but these kinds of models are data demanding and difficult to implement for
operational use over data scarce regions. We use the temperature index model as it needs
minimal data input and is computationally simple. Our results also show that the temperature
index model performs fairly well, consistent with many other studies (e.g., Li and Simonovic,
2002; Griessinger et al., 2016).

The model performance for estimating peak surface runoff $Q_{runoff}$ is shown in Table 2 and

Fig. 4B. Compared with the observed mean $Q_{runoff}$ of 1.26 mm day$^{-1}$ over the study period, the
model has a mean absolute error of $MAE$=0.1 mm day$^{-1}$, or 7.6% of the observed value. The
modelled $Q_{runoff}$ for the 12 years has a correlation coefficient of $r$=0.82 with the observed values



at a significance level of $p<0.001$. The Nash–Sutcliffe model efficiency coefficient for surface
runoff estimates is $E=0.50$, slightly lower than that for the baseflow estimates.

To further determine the relative importance of $T_a$ and $S_b$ in $Q_{runoff}$, we analysed the

relationship between $S_b$ and $Q_{runoff}$ (Fig. 5) and found that $Q_{runoff}$ showed little correlation with $S_b$.
The result indicates that total snow amount at spring breakup has little impact on the $Q_{runoff}$. As
such, the rising temperature during the snowmelt season is the main driver for determining the
$Q_{runoff}$ for the Mackenzie River basin. In contrast, the correlation between $S_b$ and $Q_{runoff}$ for the
Red River basin was found to be fairly strong (Fig. 5). Specifically, without including $T_a$, the $S_b$
by itself explained more than two thirds (the coefficient of determination $r^2=0.673$) of the
interannual variations in $Q_{runoff}$, suggesting that the major driver for $Q_{runoff}$ is $S_b$ for the Red River
basin (Wang and Russell, 2016). In another study by B.C. Ministry of Forests, Lands and Natural
Resource Operations (2012), the peak flows for the Lower Fraser River were analysed. The
Lower Fraser River is located in the latitudes between the Mackenzie River basin and the Red
River basin. It was reported that the snow factor contributes about 20-40%, and the weather
factors (mainly temperature) contribute about 60-80% to the flood risk. The above results for the
three basins are consistent and appear to suggest that the principal drivers for peak river flows
vary between basins. For northern basins temperature plays an important role, whereas for more
southern basins the amount of snow accumulation is more important.

The difference in the main drivers for $Q_{runoff}$ for the basins is largely due to the difference

in their hydroclimatic conditions. The Red River basin has a mean snow accumulation at spring
breakup of $S_b=73.3$ mm, which is less than half the accumulation for the Mackenzie River basin
($S_b=160.0$ mm). On the other hand, the Red River basin has a much larger interannual variations
of $S_b$ than that for the Mackenzie River basin. The coefficient of variation (CV) of $S_b$, or relative





standard deviation (RSD), which is calculated as the ratio of one standard deviation to the mean,
is as high as 49.2% for the Red River basin. In contrast, the CV is only 14.7% for the Mackenzie
River basin. The small $S_b$ with its large interannual variations lead to the fact that years with
large $S_b$ often correspond to severe floods, and years with small $S_b$ often correspond to very low
flows, in the Red River (Wang and Russell, 2016). For the Mackenzie River basin, large and
relatively stable snow amounts lead to the fact that the interannual variations of $Q_{runoff}$ are very
small and they are mainly determined by the temperature anomalies during the snowmelt season.
The identification of major drivers for peak surface runoff for different basins is of importance in
river flow modelling and flood forecasting.
The model performance for estimating peak river flow $Q_{peak}$, based on the above results
for $Q_{base}$ and $Q_{runoff}$, is shown in Table 3 and Fig. 4C. Compared to the mean observed $Q_{peak}$ of
1.46 mm day$^{-1}$ (28,400 km$^3$ s$^{-1}$) over the study period, the model result has a mean absolute error
of $MAE$=0.1 mm day$^{-1}$ (1,878 km$^3$ s$^{-1}$), or 6.5% of the mean $Q_{peak}$ value. The modelled $Q_{peak}$ for
the 12 years has a correlation coefficient of $r$=0.83 with the observed values at a significance
level of $p$<0.001. The Nash–Sutcliffe model efficiency coefficient for peak river flow estimates
is $E$=0.51. Of the peak river flow, 15% is contributed by baseflow and 85% by surface runoff. As
such, the modelling accuracy in $Q_{base}$ plays a small role in the modelling accuracy of peak river
flows or flood forecasts. However, modelling accuracy for total winter baseflow ($Q_{sum}$) could be
of importance as $Q_{sum}$ directly affects the estimate of $S_b$ at spring break-up (Fig. 3), which is the
case for the Red River basin. Compared to the dates for peak snowmelt, the dates for the peak
river flow observed at the station had a delay varying from 13 to 41 days among the 12 years
(Fig. 6). On average, the delay was ~22 days. The hysteresis indicates the average travel time for
the snowmelt water over the basin to reach the hydrometric station.





Results from the LOO-CV show that the 12 models trained using the 12 sets of n-1 (11
years) samples all achieved a correlation coefficient of $r>0.8$ with the observed peak river flows
at a significance level of $p<0.003$, except for the run with year 2013 left-out which had a $r=0.71$
and a significance level of $p<0.014$. The model forecasts for peak river flows based on the 12
models (Fig. 7) had a $r=0.72$ and $MAE=0.14$ mm day$^{-1}$, or 9.7% of the observed mean $Q_{peak}$
value. Compared with the model trained using data for all the years, the deterioration of the
model performance in forecasting the peak river flows from LOO-CV reflects the limited
number of data samples due to the short records of GRACE data. As the GRACE observations
continue, and with the follow-up mission of GRACE-FO, it will be necessary to recalibrate the
model to refine the parameter values and to increase the model robustness for peak river flow
estimates and flood forecasting.
The impact of measurement and leakage error in the GRACE TWS on the model results
was investigated by running the model with TWS adjusted by the error either at the pre-winter
time (for baseflow) or at the spring breakup (for surface runoff). The impact of the TWS error on
the peak river flow estimates was found to be mostly under $MAE=0.06$ mm day$^{-1}$, or 4% of the
mean $Q_{peak}$ value, which is substantially lower than the modelling error. Compared with that for
the Red River basin, the impact of the GRACE TWS error on the modelling results is much
smaller for the Mackenzie River basin. This is mainly due to the facts that for the Mackenzie
River basin (1) the error in GRACE TWS is small (see Section 2) due to its large area and (2) the
snow amount is much larger and the peak river flow is less sensitive to $S_b$ than the Red River
basin as discussed above.
The model showed a relatively lower correlation coefficient with observed peak flows for
the Mackenzie River than that for the Red River (Wang and Russell, 2016). This is not surprising



as the Mackenzie River basin is huge and very complex in physiographic and climate conditions.
Sub-basins in the Mackenzie River basin have varying flow regimes as detailed in Woo and
Thorne (2003). For instance, most of the rivers in the southern basin and at low altitudes peak in
early May, but in rivers at higher latitudes and high altitudes, where snowmelt is delayed, spring
peaks occur later. In glacierized basins, the ablation of glaciers intensifies in the summer and
this, together with snowmelt at high elevations, prolongs the high flows into summer. Large
lakes and reservoirs are highly effective in providing large storage capacities to reduce high
flows. The Mackenzie River, although it exhibits essentially a subarctic nival regime of
snowmelt-induced peak flow, is a combination of many varying flow regimes of its sub-basins.
Our model is a highly simplified representation of this complex system. In contrast, the
physiographic and climate conditions for the Red River basin are much more monotonous.
Secondly, the small interannual variations in snow amounts and peak flows of the Mackenzie
River basin increase the impact of input data errors ($Q$ and $T_a$) on the model results and impose
challenges on robust model calibration and validation.

Several water processes during the period from spring breakup to peak river flow, such as

rain, evapotranspiration, soil thaw-induced surface infiltration or groundwater recharge, as well
as lake storage and river ice dynamics, may significantly affect the magnitudes of peak river
flows. The impacts of these processes are not explicitly included in the model and could be
important sources in the modelling errors. In particular, substantial rain events over the snowmelt
period can significantly affect peak river flows. The soil thaw-induced surface infiltration and
groundwater recharge may have relatively small impacts, as the water flow is mainly from south
to north and the lower river sub-basin is usually in a frozen state when the melted water from
south arrives. Moreover, recharge of groundwater during this time period (if there is any) would



lead to an increase in baseflow, which offsets the impact of reduction in surface runoff due to
infiltration of the snowmelt water. The calibration of the model using actual peak river flows also
reduces the impact of this process on the peak river flow modelling. Nevertheless, explicitly
including the above-mentioned water processes during the period from spring breakup to peak
river flow in the model needs to be further studied.
One drawback of our method is that it does not recognise the spatial variations of snow
amounts due to the large footprint of GRACE data. Consequently, the model cannot address the
spatial snowmelt and water flow patterns within the basin which is expected to largely determine
the hysteresis between peak snowmelt and peak river flows and to further reduce the modelling
errors in peak flow forecasts. Nevertheless, as a case study with the available data and through
comparison with the results from other basins, we have demonstrated encouraging results on
peak river flow modelling and flood forecasting based only on GRACE TWS and temperature
data. In practice, our GRACE-based method can be used in combination with other available
data and methods to help improve the accuracy in peak flow or flood forecasts.
**5. Summary**
The peak river flow for the Mackenzie River is modelled in this study using GRACE
satellite observations and temperature data, which advances the applications of space-based time-
variable gravity measurements in cold region flood forecasting. The model estimates peak river
flow by simulating peak surface runoff from snowmelt and the corresponding baseflow. The
model closely estimated the observed values at a downstream hydrometric station. The results
also revealed that on average the travel time for the snowmelt water to reach the hydrometric
station is about 22 days. The major driver for determining the peak flow was found to be



atmospheric temperature variations. Compared with the Red River basin, the results highlight
that the Mackenzie River basin has relatively high water storage and water discharge capability,
and low snowmelt efficiency per unit temperature. Our GRACE-based approach for basin-scale
snowfall estimation is independent of in situ measurements and largely eliminates the limitations
and uncertainties present with traditional approaches. The results show that the GLDAS snowfall
amount in our study region is about 20% lower than our GRACE-based estimate. The model is
relatively simple and only needs data inputs of GRACE and atmospheric temperature
observations for peak flow or flood forecasting. The model can be readily applied to other cold
region basins, and could be particularly useful for regions with minimal data. In practice, this
GRACE-based method can be used in combination with other available data and methods such
as real-time flow data and flow routing models to help improve the accuracy in river flood
forecasting, and develop reservoir operation procedures for flood and water resources
management.
**Acknowledgement**

This paper was internally reviewed by Dr. Aining Zhang and Don Raymond at Canada

Centre for Mapping and Earth Observations. Comments are greatly appreciated. This study was
supported by the Groundwater Geoscience Program and the Longterm Satellite Data Records
project of the Earth Science Sector, Natural Resources Canada. GRACE land are available at
http://grace.jpl.nasa.gov, supported by the NASA MEaSUREs Program. The GLDAS data used
in this study were acquired as part of the mission of NASA's Earth Science Division and
archived and distributed by the Goddard Earth Sciences (GES) Data and Information Services
Center (DISC).



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





**Table 1. Baseflow model calibration and performance**

| $a$ ($\times 10^{-3}$ day$^{-1}$) | $b$ (mm) | Correlation ($r$) | Significance ($p<$) | Average winter baseflow (mm) | Mean Absolute Error ($MAE$) | | Nash–Sutcliffe model efficiency coefficient ($E$) |
|---|---|---|---|---|---|---|---|
| | | | | | (mm) | (%) | |
| 1.07 | -195.9 | 0.73 | 0.007 | 48.77 | 2.37 | 4.9% | 0.53 |

**Table 2. Snowmelt model calibration and model performance for peak surface runoff estimation**

| $\alpha$ (mm °C$^{-1}$) | $\beta$ (°C) | Correlation ($r$) | Significance ($p<$) | Average peak surface runoff (mm day$^{-1}$) | Mean Absolute Error ($MAE$) | | Nash–Sutcliffe model efficiency coefficient ($E$) |
|---|---|---|---|---|---|---|---|
| | | | | | (mm day$^{-1}$) | (%) | |
| 17.0 | 2.1 | 0.82 | 0.001 | 1.26 | 0.10 | 7.6 | 0.50 |

**Table 3. Model performance for peak river flow estimation**

| Correlation ($r$) | Significance ($p<$) | Average peak river flow (mm day$^{-1}$) | Mean Absolute Error ($MAE$) | | Nash–Sutcliffe model efficiency coefficient ($E$) |
|---|---|---|---|---|---|
| | | | (mm day$^{-1}$) | (%) | |
| 0.83 | 0.001 | 1.46 | 0.1 | 6.5 | 0.51 |

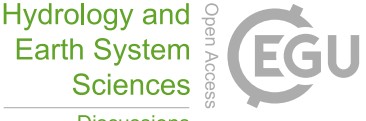

**Figure Captions:**






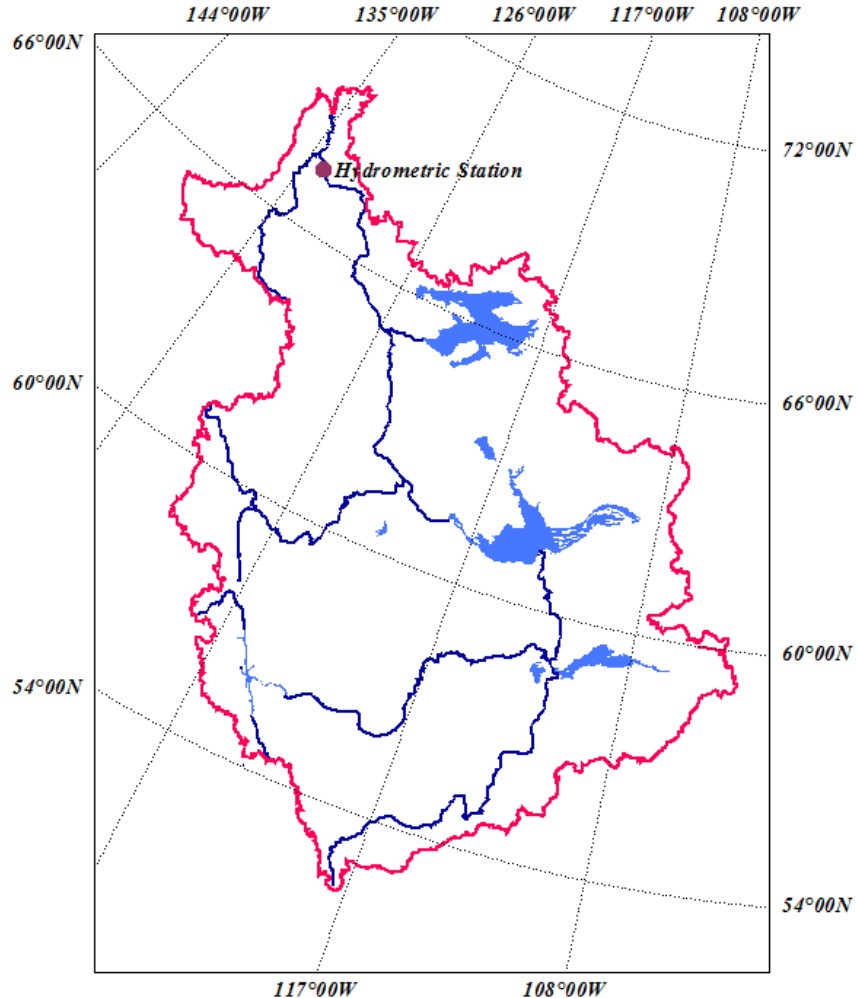


Figure 1. Map for the Mackenzie River basin and the hydrometric station used in this study.

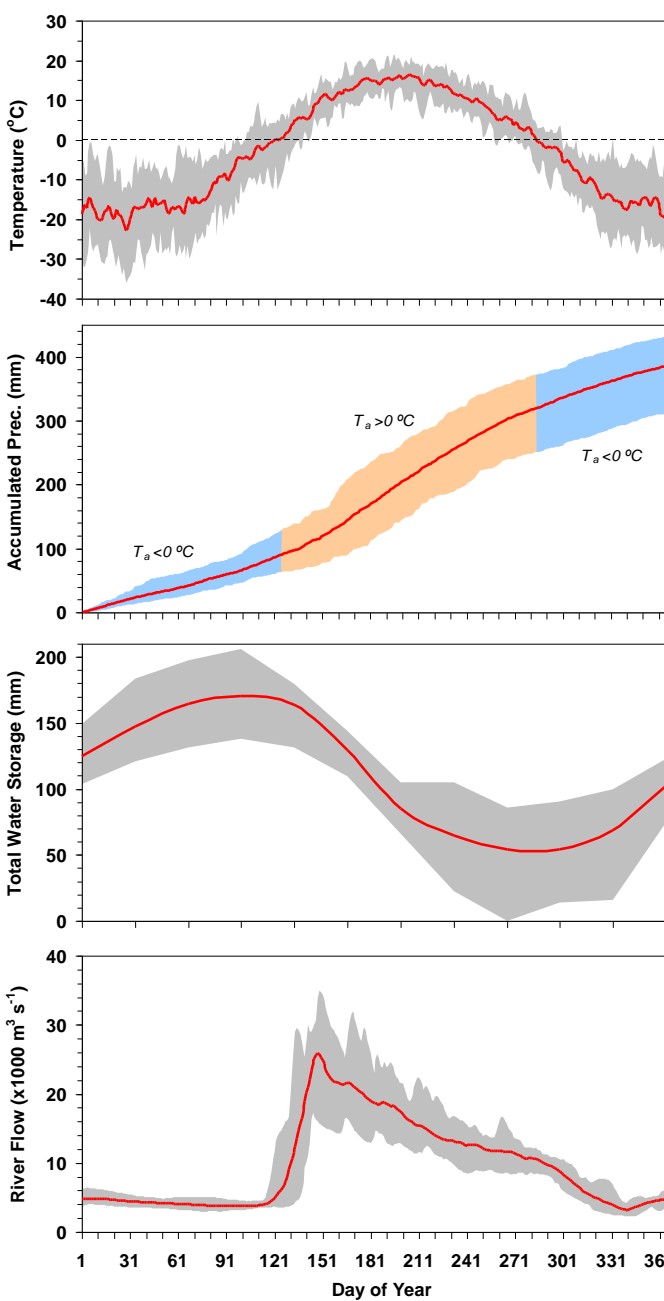


Figure 2. Hydroclimate characterization of the Mackenzie River basin (from top to bottom: air
temperature, accumulated precipitation, total water storage change, and river flow. Lines
represent the averages over the study period of 2002-2014. Shadowed areas represent the
maximum/minimum variation ranges).



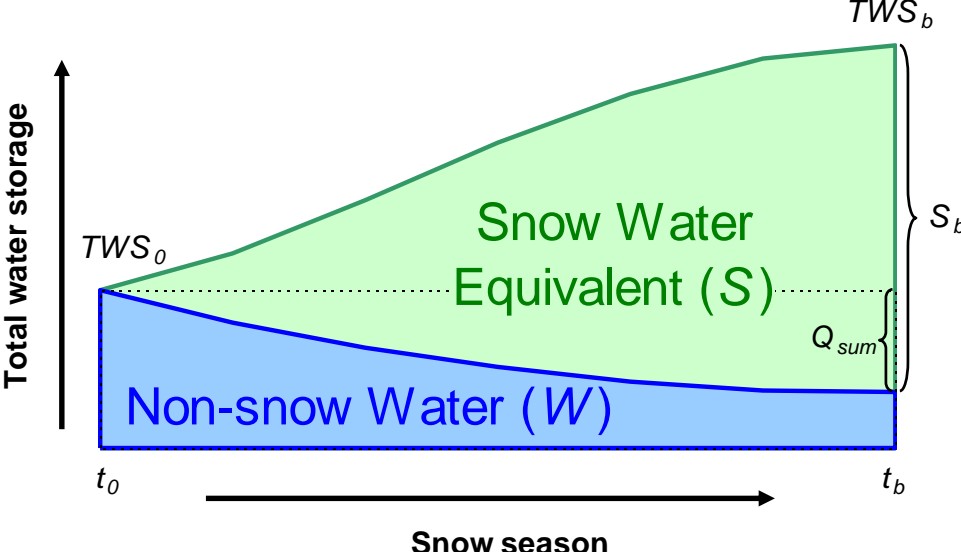


Figure 3. Diagram showing snow accumulation and non-snow water storage change in winter. ($t_0$
and $t_b$: start and breakup of snow season; $TWS_0$ and $TWS_b$: total water storage at $t_0$ and $t_b$; $W$:
non-snow water; $S_b$: Snow Water Equivalent at $t_b$; $Q_{sum}$: total river discharge in snow season).



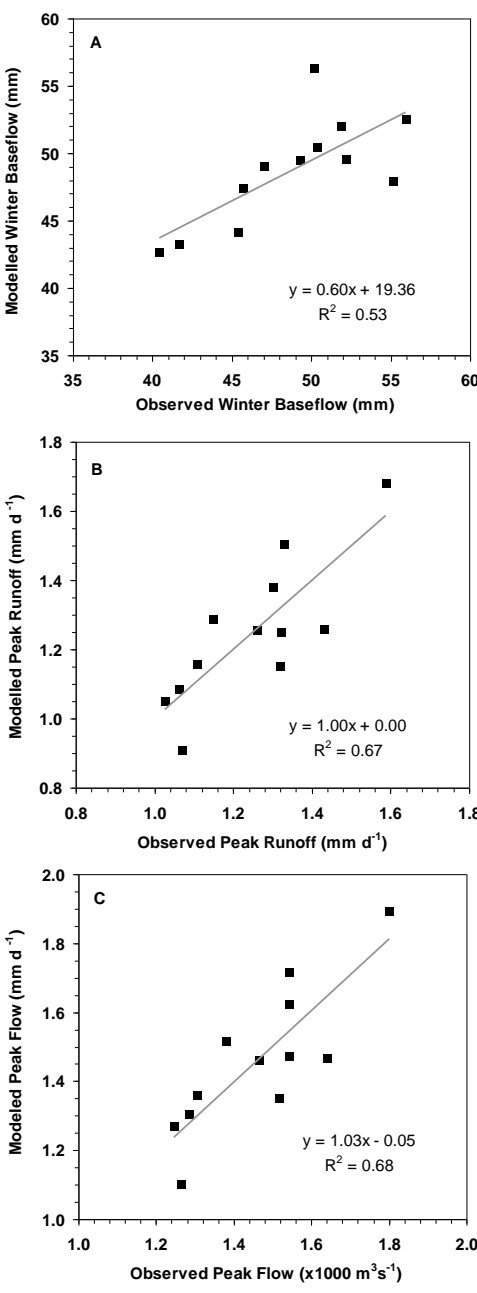


Figure 4. Comparisons of modelled vs. observed total baseflow in winter (A), peak surface
runoff (B) and peak river flow (C).



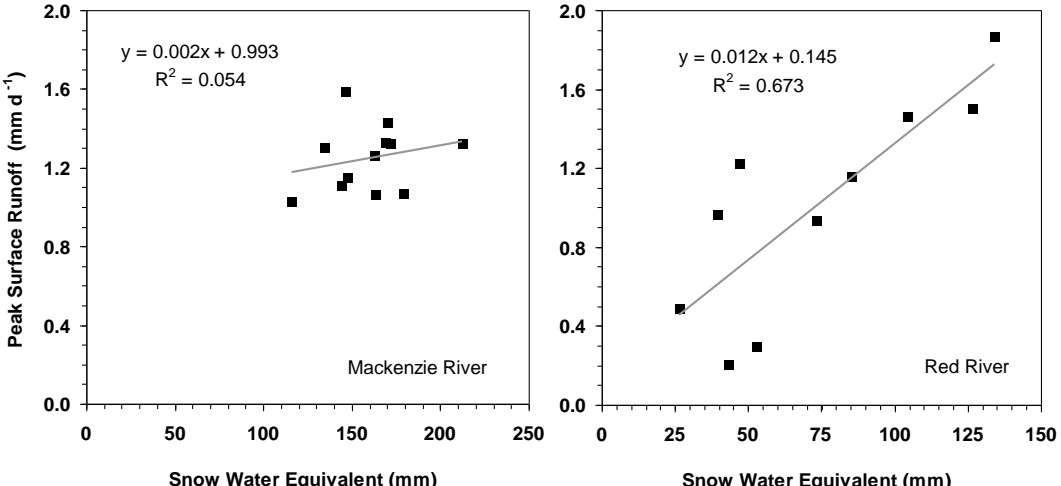


Figure 5. Peak surface runoff vs. snow water equivalent at spring break-up for the Mackenzie
River basin (left) and the Red River basin (right).







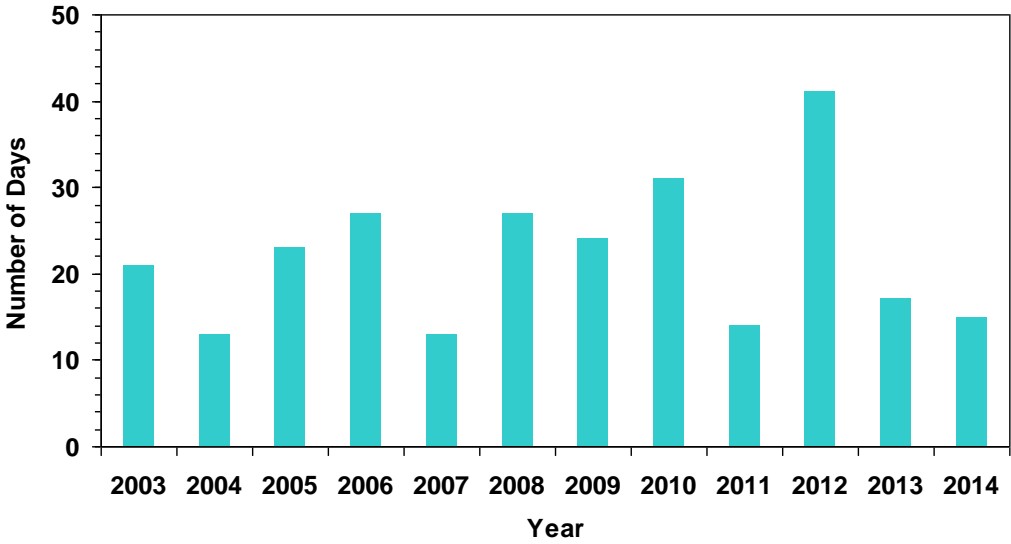


Figure 6. The travel time for snowmelt water from the basin to reach the hydrometric station.




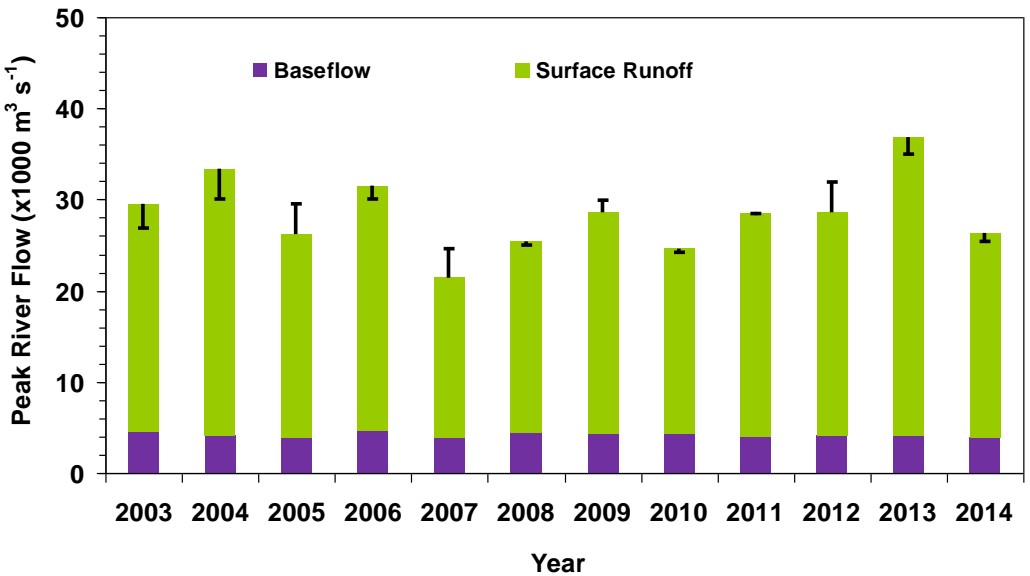


Figure 7. The leave-one-out cross-validation (LOO-CV) modelling results for peak river flow

forecast. The error bar is the difference between forecasted and observed peak river flows.