# Peer review of "Peak river flows in cold regions – Drivers and modelling using GRACE satellite"

_Hydrology and Earth System Sciences, 2016_

## Referee Comment (RC1) · Anonymous Referee #1 · 2 May 2016

This manuscript presents a modeling approach to determine the date of the peak river flow in Arctic regions. This method was applied in the McKenzie and was previously applied to the Red River (Canada). I do not know that is the meaning of determining this parameter at basin in such a large drainage basin (1.8 million km$^2$). Please find below my detailed comments:

About the introduction

Among the different technique that were used to estimate Snow Water Equivalent (SWE), you can mention that an inverse method was developed to separate SWE from Terrestrial Water Storage (TWS) estimated by the GRACE mission (Frappart et al., 2006; 2011), around line 75. Due to the principle of the GRACE mission (e.g., Schmidt

et al., 2008), there is no footprint associated to the GRACE data. It is better to mention the resolution of the mission. Replace in all the manuscript.

About the study area and datasets

Clearly separate the different aspects: 3.1 Study area, 3.2 Datasets, 3.3 Hydroclimatology This three sub-parts seem disconnected. Smooth the text. GRACE data: Add the spatial resolution of the GRACE data (∼330 km at degree 60), line 137. What is the impact of leakage error on your estimates? How are defined the scaling factors? To what I understand, applying these factors, you make the assumption that a hydrological model that may not take into account all hydrological reservoirs can be used to scale the GRACE estimates. Does this operation seem realistic in the McKenzie drainage basin in terms of hydrology (does the model simulates surface, snow and groundwater storages . . .)?

About the methods

About the model: References are missing. Are these common equations for modeling the baseflow? McKenzie drainage basin is a quite large basin. Are these relationships valid at such spatial scale. Is the breakup time occurring the same day on the whole basin ranging from 54°N to 66°N? Can this study be refined at smaller scales? At least at major sub-basins scale? About the results Break-up date are given with a great accuracy (a specific day). But GRACE-based TWS have a monthly temporal resolution. What is the impact of this temporal resolution on your estimates? Some studies derived TWS on a daily basis (e.g., Kurtenbach et al., 2012). Would it be relevant to consider these datasets if available? It would be also interesting to compare your results to the ones from global hydrological models such as WGHM, rather doing the comparison with the same approach in another basin. The presence of extensive floodplains (lines 307-310) has for consequence to delay the water in its path. How is this accounted for in your approach? If surface runoff is minimal in winter (line 310), is it because the river is frozen? In my opinion, there is no forecasting (line 405) as your approach does not

predict the peak flow. To do so, it is necessary to have access to the GRACE data one or two months before this event. I do not think this is the case. What is the amplitude of the GRACE error? (lines 406-415).

About the tables Tables 1,2,3 could be merged.

About the Figures Figure 1 should present a map of Canada, locating the McKenzie Basin.

References Frappart F., Ramillien G., Biancamaria S., Mognard N.M., Cazenave A. (2006). Evolution of highlatitude snow mass derived from the GRACE gravimetry mission (2002 -2004), Geophysical Research Letters, 33(2), L02501, doi:10.1029/2005GL024778.

Frappart F., Ramillien G., Famiglietti J.S. (2011). Water balance of the Arctic drainage system using GRACE gravimetry products, International Journal of Remote Sensing, 32(2), 431-453, doi: 10.1080/01431160903474954.

Kurtenbach E., A. Eicker, T. Mayer-Gürr, M. Holschneider, M. Hayn, M. Fuhrmann, J. Kusche (2012), Improved daily GRACE gravity field solutions using a Kalman smoother, 59–60, 39–48.

Schmidt, R., Flechtner, F., Meyer, U., Neumayer, K. H., Dahle, C., König, R., & Kusche, J. (2008). Hydrological signals observed by the GRACE satellites. Surveys in Geophysics, 29(4-5), 319-334.

---

## Referee Comment (RC2) · Anonymous Referee #2 · 3 Jun 2016

This study provides a simple yet effective approach to spring snowmelt prediction in cold regions by incorporating temperature data and GRACE satellite observations on total water storage. The mode is applied to forecast spring peak river flow in the Mackenzie River Basin, and leads to a few very interesting results and conclusions. The approach is of great usefulness in flood prediction and water resources management in cold regions where data is scarce, such as the Mackenzie River Basin. The paper is also very well organized and presented. I highly recommend for publication in HESS. I have only a few minor questions and comments as listed below:

1) There are a couple of typos: Line 255 "is" -> "are"; Line 383&384 "km3" ->"m3"

2) Add the number of samples (n) in Tables 1 and 2 for the calibration of the two submodels, the Baseflow model and the Snowmelt model.

3) Looking at Fig. 3 and Eq. (3), my understand is: TWS0 is the total terrestrial water storage at the beginning of a snow season when non-snow water (W) is the largest, and the base flow (Qbase) observed at the hydrometric station during the snow season is modelled using Eq(3). Given a positive value of parameter a and a negative value of parameter b (-195.9 mm; Lines 303-304), this means that the observed base flow is larger than 0 even if TWS0<0. Then, it seems that TWS0 does not only reveal "non-snow water" but also something else.

4) Eq. (5) provides a nonlinear model for determination of snow melt rate using the snow amount and the temperature; Fig. 5a shows a weak correlation between SWE at spring break-up and peak surface run-off for Mackenzie Bain, therefore temperature is considered to have played an important role (Lines 353-355). I agree with the authors that a large capacity of retaining and releasing no-snow water by the basin during a year's cycle would explain this. It would be of great interest to exploit and understand how the temperature impacts the annual variability of peak river flow, and the days of delays from peak snow melt to peak river flow observed at the hydrometric station; but this might not be relevant to the current paper.

---

## Author Comment (AC1) · 7 Jun 2016

We greatly appreciate the comments which helped improve the paper. The main objectives of this paper are (1) to test the model performance of Wang and Russell (2016) in forecasting the magnitudes of peak flows for the Mackenzie River and (2) to identify the differences in major drivers for determining the peak flows between the Mackenzie River and Red River. The date for peak river flow in the study is from in situ observations, and it is used to estimate the overall water travel time by comparing it with the modelled date for peak snowmelt.

We added a brief summary for the studies by Frappart et al. (2006, 2011) for estimating SWE from the GRACE TWS in the Introduction section, and replaced "footprint" by

"resolution" throughout the manuscript.

Section 2 is separated into three subsections: 2.1 Study Basin, 2.2 Datasets, and 2.3 Basin Hydroclimatology, to make the content better presented. More details about the GRACE data, including spatial resolution and the definition and application of scaling factors, are added. The leakage error in TWS, which is estimated at less than 10 mm of water, or less than 9% of the average seasonal variation of TWS, is relatively small. The impact of leakage error on the peak river flow estimates is evaluated and discussed on Page 20 (revised version). Overall, the impact was found to be small (MAE mostly under 0.06 mm day-1, or 4% of the mean peak flow value). The scaling factors were based on NCAR's land surface model CLM4. It includes simulation for surface water, soil water and snow, but not groundwater storage in aquifers. The lack of groundwater components and the uncertainties in CLM outputs affect the effectiveness of the scaling process. Fortunately, the hydrology of the basin during the study season (winter-early spring) is primarily determined by the snow processes, while groundwater variation is relatively small.

The missing reference for the method is added. The baseflow model we used is a modified Linear Reservoir model which is widely used in baseflow recession analysis. It is difficult to construct physically-based baseflow models for the basin due to very limited data and knowledge for the basin hydrogeology. Also, since baseflow contribution to the flood is very small in our case, the impact of its estimation uncertainty from our model on flood forecasting is minor. Our model doesn't explicitly include the physical details of a basin. High spatial heterogeneities within a basin (e.g., difference in breakup time for a large basin) would likely increase uncertainties in estimates. We investigated the scale impact on model performance by comparing the results for the Mackenzie River basin (large) and the Red River basin (small). As expected, the model performs better for the smaller basin which is more homogeneous in climate (including breakup time), land surface, and hydrogeological conditions. More discussions on this have been added in the paper. We selected Red River basin instead of a sub-basin

Interactive
comment

within Mackenzie River basin as (1) Red River basin has large differences in hydroclimate with that in Mackenzie River basin. By comparing results from them we can better reveal the differences in major drivers for the peak river flows over different regions, which is a major objective of this study; and (2) Red River basin is more homogeneous with better observations. However, it is worth noting that even for the small basin of Red River (whose size is at the lower limit for GRACE resolution), the size is still large enough to have significant spatial differences in snow breakup dates (see discussion in Wang and Russell, 2016). Moreover, small basins tend to have large uncertainties in the TWS estimates due to the coarse resolution of GRACE satellites. This can be clearly seen by the large difference of impacts of the leakage and measurement errors in the TWS on the flood estimates as discussed in the paper. The paper has been revised by adding clarifications and more discussion on this.

The break-up and peak snowmelt dates are obtained in the snowmelt model which has a daily time step. The GRACE data is only used to estimate the initial condition (TWE) for the snowmelt model. As the snow pack accumulation is a rather smooth process, the impact of monthly temporal resolution of GRACE data on the model estimates is largely reduced. However, it is agreed that GRACE TWS derived on a daily basis would better fit the modelling scheme and improve the model estimates. In particular, we believe that GRACE TWS daily data will have large potentials for many other applications such as estimating storm-induced flood, soil moisture, and snowmelt process. We compared the results from the same model over different basins as one main objective of the study is to identify the differences in major drivers for determining the peak flows between different basins, with an intention to provide a simple approach to complement the existing operational flood forecasting approaches by using additional data from GRACE. The flood in our case is mainly a result of snowmelt. To our understanding, WGHM uses degree-day for snow simulation and it assumes that snow melts with a constant rate when temperature is above $0°C$ (2 mm/day per degree in forests and 4 mm/day in other land cover types). Our snowmelt model uses similar algorithm to that in WGHM, but its parameters (e.g., melt rate) are estimated based on observation

data in the basin. As such, we expect our results better reflect the snowmelt process in the basins. For more detailed hydrological studies, we have developed the land surface model EALCO which has comprehensive algorithms for snow simulation such as dynamic snow layering and fractional cover algorithms, dynamic snow albedo, snow cover compaction, destructive metamorphisms, and the implicit solutions of heat and melt water transfer equations in the snow pack (Zhang et al., 2008). However, comprehensive hydrological or land surface models demand a large number of data inputs and are more difficult to be used for operational flood forecasting. Compared with dry land or some other lands such as forests, basins with wetland and water bodies have higher water dischargeability, as shown in our results. This is accounted in our approach by the parameter (a) fitting using observed Q. Surface runoff is minimal in winter as there is basically no liquid precipitation during the season. The river flow in winter is mainly associated with groundwater and water body discharges. The model forecasting capability relies on the snowmelt model using temperature input, while GRACE provides the initial condition (SWE) at the breakup, so it is not necessary to have GRACE data that is months in advance. However, it is worth mentioning that GRACE data in the epoch that is a couple of months before the spring breakup might still be very useful for freshet flood warning, as snow packs have high correlations between the two epochs in our study region (e.g., February vs. April). The GRACE TWS error is estimated at 9.9 mm, which is about 8.5 % of the average seasonal variation of TWS (see Section 2.3). It is estimated following the approach of Wahr et al. (2006) (see Section 2.2). The paper has been revised by adding more discussion on this. The reference of Kurtenbach et al. (2012) is added.

Tables 1, 2, and 3 in the previous version are merged as suggested.

Figure 1 is revised by adding the map for Canada and locating Mackenzie River basin in the map (attached below), as suggested.

References: Wang, S. and Russell, H. 2016. Forecasting snowmelt-induced flooding using GRACE satellite data: A case study for the Red River watershed. Canadian

Journal of Remote Sensing 42, 1-11, DOI: 10.1080/07038992.2016.1171134.

Zhang, Y., Wang, S., Barr, A.G., and Black, T.A. 2008. Impact of snow cover on soil temperature and its simulation in the EALCO model. Cold Regions Science and Technology, 52, 355-370.

Revised Figure 1.
* * *
[Figure]

Fig. 1. Map for the Mackenzie River basin and the hydrometric station used in this study. The Red River basin is shown in the map for Canada.

---

## Author Comment (AC2) · 7 Jun 2016

We greatly appreciate the comments from the reviewer. Response to each comment is listed below.

1) The typos have been corrected. Yes, the water flow unit should be m3 s-1.

2) The number of samples used for the model development and test is added in the tables.

3) The TWS from GRACE are values relative to a reference (baseline) value. The baseline in the original TWS data was based on the average from January 2004 through December 2009. In this study, it was adjusted to the minimum value found over the

12-year study period, as described in Section 2.2. So, when TWS<0, it means that the basin total water storage is below this reference level, under which a basin may still be possible to discharge water. This is the case for the Mackenzie River basin as shown by both the observational data and the negative value of parameter b fitted for the model.

4) To explore and understand the major drivers for determining the interannual variations of peak flows is one of the major objectives of this paper. What we found is that temperature variations during the snowmelt season is the main driver for determining the interannual variability of peak river flows for the Mackenzie River basin. The interannual variations of the days from peak snowmelt to peak river flow at the hydrometric station could be a result of several factors, including the magnitudes of flows for the days around the peak flow date and the locations of snowmelt events that mainly contribute to the peak flow, etc. We agree with the reviewer that understanding the travel time for snowmelt water to reach a certain point of the river is important. It likely needs a spatially distributed water routing model and it is beyond the scope of this study.